

# Hybrid purity identification using EST-SSR markers and heterosis analysis of quantitative traits of Russian wildrye

Zhiqi Gao[1,*], Lan Yun[1,2,*], Zhen Li[1], Qiyu Liu[1], Chen Zhang[3], Yingmei Ma[4] and Fengling Shi[1]

[1] College of Grassland, Resources and Environment, Inner Mongolia Agricultural University, Hohhot, China
[2] Ministry of Education Key Laboratory of Grassland Resources, Hohhot, China
[3] College of Animal Science and Technology, Hebei North University, Zhangjiakou, China
[4] College of Desert Control Science and Engineering, Inner Mongolia Agricultural University, Hohhot, China
* These authors contributed equally to this work.

Corresponding author
Lan Yun, yunlan@imau.edu.cn

## ABSTRACT

Russian wildrye, *Psathyrostachys junceus* (Fisch.) Nevski, is widely distributed in the high latitude areas of Eurasia. It plays an important role in grassland ecosystem maintenance, as well as being a valuable palatable forage species for livestock and wildlife. Russian wildrye germplasm has rich phenotypic and genetic diversity and has potential for improvement through crossbreeding. In this study, fifteen Russian wildrye hybrid combinations were produced and one F1 population with 123 putative hybrids was obtained by crossing two individual plants with significant differences in nutritional characteristics and reproductive tiller number. Twelve phenotypic traits of the F1 population were measured for three consecutive years, and ten of the twelve traits were in line with the genetic characteristics of quantitative traits. Hybrid superiority was revealed among F1 hybrids in both nutritional and reproductive traits. One non-recurrent parent plant with the highest PCA-synthesis score was selected and used to make a backcross with the 'BOZOISKY SELECT' male parent, and 143 putative BC1 hybrids were obtained. Sixteen pairs of EST-SSR primers were randomly selected from polymorphic primers derived from different expressed tiller trait related genes. Three primer pairs that amplified both the paternal and maternal characteristic band were used to assess the purity of the F1 population, and three primer pairs (with one shared primer pair) were used to identify the BC1 population. The hybrid purity was 96.75% for the F1 population and 95.80% for the BC1 population, and the results were confirmed by self-fertility test through bagging isolation. The genetic similarity coefficients between the F1 progeny and the male parent ranged from 0.500 to 0.895, and those between the BC1 progeny and the male parent ranged from 0.667 to 0.939. A subset of individuals in the BC1 population had closer genetic distance to the recurrent parent, and genetic variation within the BC1 population decreased compared to the F1 population.

## INTRODUCTION

Russian wildrye (*Psathyrostachys juncea*) is a perennial cross-pollinated bunch grass species of *Gramineae* (Triticeae) with a diploid chromosome number of 2n = 14 (*Wei, Campbell & Wang, 1997*). The wild Russian wildrye germplasm originated in Siberia and was introduced and popularized in western Canada as pasture plants in the 1830s (*Dormaar et al., 1995*), and the first cultivated variety was released by the USDA in Mandan, North Dakota, in 1927 (*Rogler, 1963*). In China, this species is mainly distributed in the north of the Tianshan Mountains in Xinjiang, the mid-west of Inner Mongolia and in Tibet and has been widely used for grassland restoration in these areas. Russian wildrye is a high-quality forage that is well adapted to the semiarid and arid rangelands of North America and China, in part due to its strong tolerance to drought and moderate tolerance to saline-alkali soil (*Asay, 1992*; *Asay & Jensen, 1996*). It has a strong root system with short rhizomes and dense fibrous roots that enables Russian wildrye to compete for limited available water and enhances its drought tolerance (*Sbatella et al., 2011*). Russian wildrye has high feed value for livestock and wildlife due to its good palatability. However, its relatively poor forage and seed yield suggest that Russian wildrye needs to be improved to meet users' production objectives. A germplasm evaluation trial showed that significant variability exists between accessions from different regions (*Bai, 2016*), and showed significant breeding potential for forage and seed yield improvement.

Hybridization combines parents' genes and provides materials with high genetic heterozygosity for selection (*Han et al., 2020*; *Fiévet et al., 2018*). Heterosis refers to the phenomenon whereby the first generation of hybrids produced by crossbreeding of parents with different genotypes is superior to parents in one or more traits, such as growth potential, viability, fecundity, stress resistance, yield or other quality traits (*Ray et al., 2021*). Hybrid breeding and heterosis has been exploited in perennial ryegrass and tall fescue and has demonstrated its superior performance over other breeding strategies (*Vogt et al., 2020*; *Kindiger, 2021*). A number of experiments have been performed to apply hybrid breeding to self-incompatible perennial forages grasses and found that heterosis has potential to improve forage yield, ratooning ability, disease resistance and forage digestibility of forage grasses, such as Japanese timothy (*Phlieum pratense* L.) (*Tamaki et al., 2009*; *Tanaka et al., 2013*) and Bermuda grass (*Cynodon dactylon* (L.) pers.) variety 'Coast Cross-1' (*Burton & Monson, 1972*; *Taliaferro et al., 2002*). Analysis of the hybrid traits and heterosis potential of Russian wildrye is critical for further developing breeding programs. Russian wildrye is a perennial forage grass with a low ploidy level, self-incompatibility and cross-pollination, and is highly recommended for hybrid breeding. In 2003, the Canadian Centre for Semi-Arid Grassland Agricultural Research developed a Russian wildrye variety 'Tom', which had significantly higher forage yield than its parents (*McLeod et al., 2003*). *Jensen et al. (2005)* crossed diploid and tetraploid Russian wildrye accessions, and improved the hybrid offspring's viability, seed yield and dry matter yield. The USDA-ARS and the Agricultural Experimental Station issued and promoted a new variety 'Bozoisky-II' with excellent seedling vigor, seed quality and yield, dry matter yield and drought resistance, which was bred using multiple groups of parent
combinations of Russian wildrye populations and multiple hybridizations (*Jensen et al., 2006*). The identification and demarcation of heterotic groups are essential for successful hybrid strategy in most species.

Selecting suitable parents for hybridization is the basis of a successful breeding program (*Nair & Shylaraj, 2021*). In this research, an F1 population with the best field performance and good fertility was obtained after a hybridization trial of different Russian wildrye germplasms, with the female parent plant selected from wild germplasm from Xinjiang, China, and the male parent plant selected from the 'BOZOISKY SELECT' variety introduced from North America, which implies remarkable geographical differences between parent plants. In recent years, the use of molecular markers for hybrid identification has been widely applied in perennial forage grass (*Ji et al., 2009*; *Xie, Zhang & Cheng, 2010*; *Wang et al., 2021*). EST-SSR markers have a high amplification success rate and related gene annotation rate since they come from the transcription region of genome DNA (*Sun et al., 2021*). Russian wildrye is a non-model plant species with little genomic information, and screening EST-SSR markers to distinguish genotypes of hybrid progeny is an essential hybrid breeding procedure.

This study aimed to develop an EST sequence library and screen SSR markers that can be used to distinguish Russian wildrye hybrid offspring genotypes, evaluate heterosis and breeding potential of Russian wildrye hybrids, and to select F1 individuals that are dominant in target traits for backcrossing and to introgress target traits into the backcross population.

## MATERIALS AND METHOD

### Plant materials

Eighteen Russian wildrye accessions from different countries and regions were collected and planted in a greenhouse and transplanted to the field individually in August of the same year. Thirty to forty plants were transplanted for each accession. All plant germplasms used in this research were provided by the National Plant Germplasm System, U.S.A., and the National Medium-term Gene Bank of Forage Germplasm, China. The accession numbers, geographical origins and germplasm preservation institutions of all materials are summarized in Table S1.

### Hybridization and combination selection

Tiller height (cm), reproductive tiller number, nutritional tiller number, and spike length (cm) traits of 18 Russian wildrye accessions were measured at heading period. Fifteen hybrid combinations were assembled based on the phenotype and genetic distance between individuals estimated by *Zhang et al. (2019)*. Geographically distant accessions were selected if two accessions had the same genetic distance. Accessions with advantageous traits were selected as male parents on the complementary traits. Furthermore, the flowering time of both parents must be compatible.

After determining the hybrid combinations, the female parents were bag-isolated before flowering. The male pollen was collected at full flowering stage for artificial bagging pollination, and each combination was pollinated three to five times. The hybrid seeds of

each female parent were harvested. Combinations with high seed setting rate and hybrid populations with sufficient offspring (more than 100 plants) were selected for hybrid identification, evaluation and subsequent backcrossing. Hybrid seeds were planted in a greenhouse and seedlings were raised in October. Hybrid plants were transplanted to the experimental plot in May of the next year with planting spacing of 50 cm × 50 cm. The experiment was conducted in the forage experiment station of Inner Mongolia Agricultural University, which is located in Saihan District, Hohhot, Inner Mongolia, China (111.41°E, 40.48°N). The hybrid progeny evaluation was conducted in the flowering seasons during three consecutive years of field growth.

## EST-SSR primers design

Based on Sequencing by Synthesis (SBS) technology, the Illumina Hiseq high-throughput Sequencing platform was used to sequence six cDNA libraries of mixed shoot meristem RNA of seventeen Russian wildrye accessions. In total, 400 SSR loci were screened from significantly differently expressed genes between dense and sparse tiller groups. One hundred and three EST-SSR primer pairs were selected after polymorphism test. The methods used have been described in *Li et al. (2022)*. The sequence data was deposited in the NCBI SRA (Sequence read archive) database (https://www.ncbi.nlm.nih.gov/sra/PRJNA789128) (Accession numbers: SRR17509835, SRR17509834, SRR17509833, SRR17509832, SRR17509831, SRR17509830). The sixteen pairs of EST-SSR primers used in the experiment were randomly selected from the 103 pairs of polymorphic primers developed and used in Russian wildrye genetic structure analysis by *Li et al. (2022)* and synthesized by Sangon Biotech (Shanghai, China).

## DNA extraction, PCR amplification and marker validation

Genomic DNA was extracted from 123 F1 individual plants, 143 BC1 individual plants and parents. The leaves were wiped with alcohol and deionized water and 1 g of leaves per plant was cut and brought to the laboratory and stored at −80 °C. DNA was extracted using a plant genome DNA extraction kit (Tiangen Biotech, Co., Ltd, Beijing, China) and stored at −80 °C. The quality and quantity of DNA was evaluated on 1% agarose gel with a NanoDrop2000. The DNA concentration was adjusted to 50 ng/μl for SSR marker analysis.

PCR was performed in a total volume of 15 μl containing 50 ng of genomic DNA, 7.5 μl 2×Taq PCR Mastermix (including 0.1U of Taq DNA polymerase, 3.0 mM MgCl$_2$, 500 μM of dNTPs, 100 mM KCl, 20 mM Tris-HCl; Tiangen Biotech, Co., Ltd, Beijing, China), 4.5 μl ddH$_2$O, and 1 μl of each forward and reverse primer. PCR amplification was performed using Applied Biosystems (Life Technologies Holdings Pte. Ltd, Singapore) with the following cycling conditions: pre-denaturation at 95 °C for 4 min followed by 33 cycles of 95 °C for 45 s, 54.7 °C to 57.5 °C (depending on primers) for 30 s and 72 °C for 30 s, and finally 10 min extension at 72 °C. PCR products were detected using 6.0% polyacrylamide gel electrophoresis (PAGE) and gel-visualized by silver staining. The 50 bp DNA ladder and 100 bp DNA ladder (Tiangen Biotech, Co., Ltd, Beijing, China) were used as the standard size markers.

## EST-SSR marker analysis and hybrid purity identification

EST-SSR primers with specific loci between male and female parents were selected to identify the authenticity of the progenies of the 123 F1 and 143 BC1 Russian wildrye.

The observed number of alleles, effective number of alleles, *Nei's (1973)* gene diversity, Shannon's Information index, and the percentage of polymorphic bands for each of the genetic SSR markers were calculated using Popgene32. The genetic similarity coefficient was calculated using Ntsys-2 (version 2.10) software. Hybrid population purity was measured as:

$$
\begin{aligned}
\text{Hybrid purity }(\%) = (\text{number of tested plants } - \\
\text{number of plants with female marker type})/ \\
\text{number of tested plants} \times 100\%.
\end{aligned}
$$

Five single spikes were selected from the female parent plant for bagging isolation, and hybrid purity was further verified by measuring the self-fertility rate of the female parent. The self-fertility rate of five single plants and five single spikes per plant of the F1 population were bag isolated and compared with the female parent single spike to determine whether there were differences between them. The self-fertility rate was calculated as:

$$
\text{Self-fertility rate }(\%) = \text{seed number per spike/spikelet number per spike} \times 100\%.
$$

## Frequency distribution and correlation analysis of phenotypic traits of the F1 population

After the population had been established, phenotypic traits of the F1 population were measured at the blooming stage for three consecutive years. All traits were measured with five repetitions of each single plant each year. The phenotypic traits measured included: tiller height (cm), second leaf length (cm), second leaf width (cm), reproductive tiller number, nutritional tiller number, spike length (cm), spike width (cm), spikelet number per spike, seed number per spike, seed weight per spike (g), seed number per plant and thousand kernel weight (g). Normality tests of the F1 population phenotypic traits and frequency distribution analysis were conducted using Origin 2019b (MicroCal, Northampton, MA, USA). After selecting the appropriate correlation coefficient according to the distribution, correlation analysis of phenotypic traits was implemented in R 4.0.5 (https://cran.r-project.org/bin/windows/base/old/4.0.5/).

## Heterosis analysis of the F1 population

The coefficient of variation, mid-parent heterosis and heterobeltiosis were used to evaluate the separation level of F1 population traits and the field performance of hybrids. The calculation formulae were (*Sun et al., 2013*):

$$
\text{Coefficient of variation }(CV) = \text{Standard deviation }(SD)/\text{Mean}
$$

**Table 1 Progeny performance of each hybrid combination of *Psathyrostachys juncea*.**

| Combinations (♀ × ♂) | Nutritional tiller number (♀/♂) | Reproductive tiller number (♀/♂) | Spike length (♀/♂) (cm) | Number of offspring seeds harvested | Survival rate (%) | Overwintering rate (%) | Proportion of fertile plants (%) |
|---|---|---|---|---|---|---|---|
| 502,573 × 565,051 | 22/113 | 76/130 | 15.60/15.80 | 54 | 3.70% | 50.00% | 100.00% |
| 502,576 × 272,136 | 46/120 | 11/95 | 12.80/12.30 | 30 | — | — | — |
| 531,826 × 476,299 | 49/67 | 73/68 | 14.00/11.90 | 19 | — | — | — |
| 619,483 × 565,060 | 29/102 | 12/34 | 10.80/13.20 | 8 | — | — | — |
| 565,052 × 565,051 | 31/73 | 37/51 | 12.12/16.13 | 4 | — | — | — |
| 502,577 × 565,051 | 34/113 | 6/130 | 12.96/15.80 | 9 | — | — | — |
| 406,468 × BOZOISKY SELECT | 33/142 | 23/94 | 10.81/12.02 | 29 | — | — | — |
| 598,610 × 476,299 | 35/67 | 15/68 | 10.98/11.90 | 8 | — | — | — |
| 598,610 × 565,044 | 32/58 | 11/20 | 10.86/14.36 | 50 | 4.00% | 50.00% | 0.00% |
| 598,610 × BOZOISKY SELECT | 40/55 | 12/139 | 15.61/11.18 | 46 | — | — | — |
| 595,135 × 476,299 | 78/74 | 3/35 | 15.50/11.38 | 216 | 31.48% | 44.12% | 16.67% |
| XJ-ALT × BOZOISKY SELECT | 33/87 | 28/113 | 15.47/13.31 | 300 | 49.00% | 83.67% | 95.12% |
| 598,611 × 476,299 | 17/67 | 7/68 | 11.37/11.85 | 59 | 0.00% | 0.00% | 0.00% |
| 598,611 × 565,044 | 30/58 | 14/20 | 12.20/14.36 | 16 | — | — | — |
| 598,611 × BOZOISKY SELECT | 45/100 | 17/140 | 11.95/12.03 | 10 | — | — | — |

Mid-parent heterosis $= (F1\text{-}MP)/MP$,

Heterobeltiosis $= (F1\text{-}HP)/HP$,

where F1 is the mean value of a certain quantitative trait in the hybrid population, MP is the mean value of the same trait in the parents, and HP is the higher value of the same trait in the parents.

# RESULTS

## Selection of hybrid combinations

In total, fifteen hybrid combinations were produced by control pollination. The parents in each hybrid combination showed great differences in nutritional and reproductive tiller traits, and most of them were genetically and geographically distant. The traits of parents in each hybrid combination and the survival and development of progeny plants were statistically analyzed (Table 1). Combinations with high seed setting rate were screened. Five hybrid combinations harvested more than 50 hybrid seeds each, and the seeds were germinated and cultivated in a greenhouse. The population was transplanted to the experimental plot and established after winter. One F1 Russian wildrye population with 123 putative hybrids was obtained by crossing a single plant selected from Russian wildrye 'XJ-ALT', a wild germplasm from Xinjiang, China, conserved in the National

Medium-term Gene Bank of Forage Germplasm, China, as a female parent, and a single plant selected from the Russian wildrye variety 'BOZOISKY SELECT' (*Asay et al., 1985*), and a germplasm from NPGS, (Bethesda, MD, USA) as a male parent. This population was used for hybrid identification, evaluation and subsequent backcrossing.

## Normality test and frequency distribution of phenotypic traits in the hybrid progeny population

For the phenotypic traits analysis of the F1 progeny, the following materials were excluded: false hybrids (No. 43, 57, 86, 105), and true hybrid plants with incomplete data due to lack of reproductive organs (No. 11, 28, 36, 64, 65, 70, 76, 77, 82, 92, 93, 96, 102, 112). According to observations of twelve traits in the F1 population, the Kolmogorov-Smirnov normality test was performed on each group of trait data and P-P plots was produced (Fig. 1). The cumulative frequency of the sample was taken as the abscissa and the sample value was taken as the ordinate. If the data fit a normal distribution, the points would be distributed around the diagonal of the first quadrant. Frequency distribution maps were made for each trait, with the sample value as the abscissa and the number of samples as the ordinate. The *p* values of ten of the twelve traits were all greater than 0.05, except for reproductive tiller number and seed number per plant. These results showed that the observation values of ten traits in this population basically followed a normal distribution. Calculating the skewness and kurtosis of the distribution curve, the twelve traits were all skewed to some degree. Ten of the twelve agronomic traits were in line with the genetic characteristics of quantitative traits in this Russian wildrye F1 hybrid population. The distribution of all twelve phenotypic traits in this population was relatively stable over 3 years (Table S2).

## Correlation analysis of agronomic traits in the hybrid population

The Pearson correlation coefficient is suitable for two continuous variables that are normally distributed. Pairwise correlation analysis of all agronomic traits (Fig. 2) showed that nutritional tiller number as an important indicator of forage yield was only significantly positively correlated with second leaf length (r = 0.415, $p \leq 0.05$). Spikelet number per spike, reproductive tiller number, spike length and seed number per plant were positively and highly significantly correlated with plant tiller height ($p \leq 0.01$), and the spikelet number per spike had the highest correlation with plant tiller height (r = 0.618). For the traits related to seed yield components, reproductive tiller number and spikelet number per spike were all significantly positively correlated with seed number per spike ($p \leq 0.01$). The correlation level between reproductive tiller number and seed number per spike was highest (r = 0.900).

## Hybrid superiority of the F1 population

Among the twelve phenotypic traits, tiller height, nutritional tiller number, second leaf length, and second leaf width could be regarded as nutritional traits. Reproductive tiller number, spike length, spike width, spikelet number per spike, seed number per spike, seed number per plant, seed weight per spike and thousand kernel weight could be regarded as

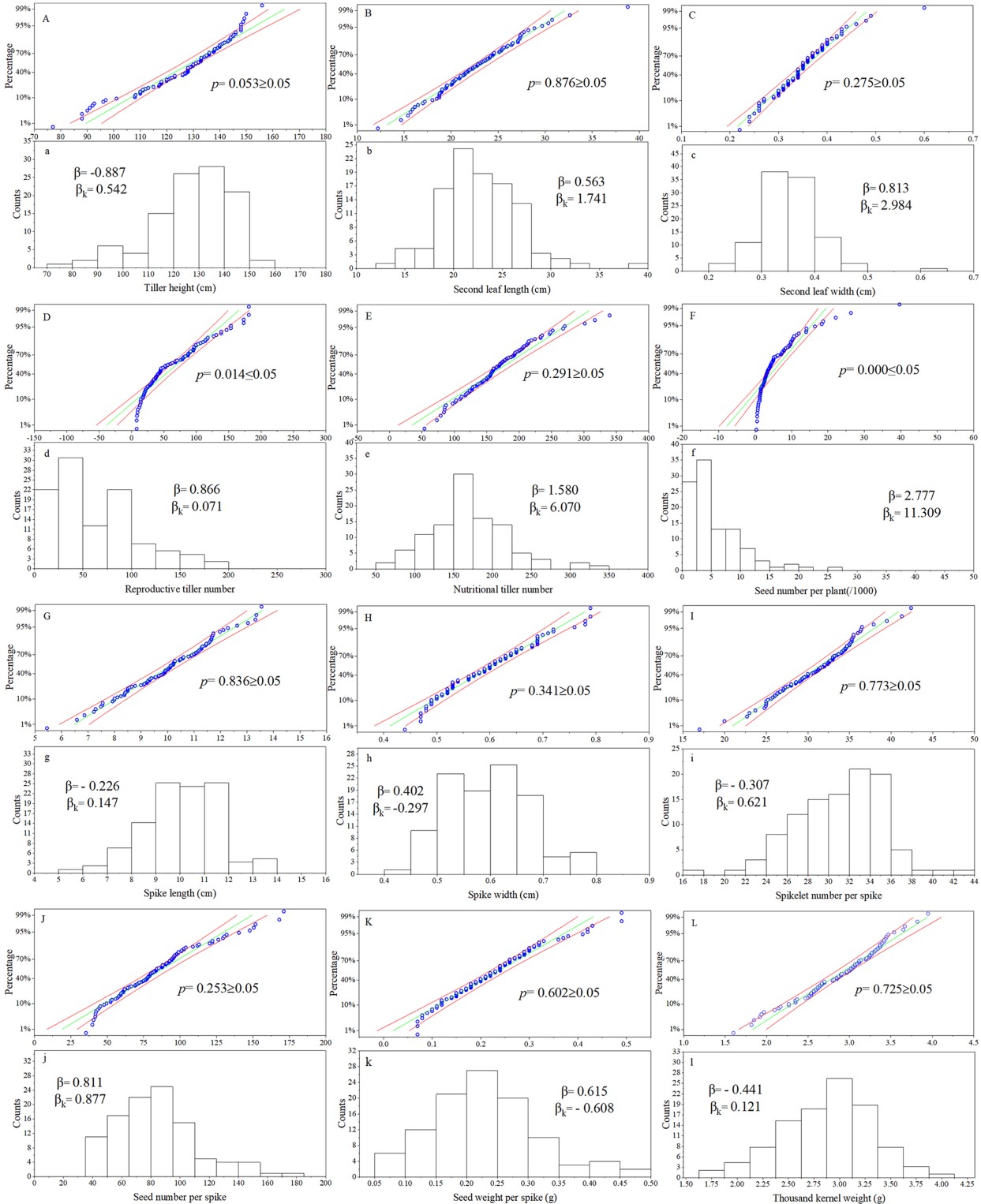

**Figure 1 P-P chart and frequency distribution histogram of agronomic traits.** (A–L) are the P-P chart of normality tests; (a–l) are frequency distribution histograms; the same letter corresponds to the same trait, such as (A) and (a) chart represents normality test and frequency distribution chart of plant height; $p$ = Kolmogorov-Smirnov normality test, $p \geq 0.05$ = at 0.05 level, this group of data is significant from the normal distribution population; $\beta$ = Skewness, describe the direction and degree of distribution asymmetry; $\beta \approx 0$, the distribution is symmetrical and obeys the normal distribution; $\beta > 0$, the distribution is right skewness, also known as positive skewness; $\beta < 0$, the distribution is left skewness, also known as negative skewness; $\beta_K$ = Kurtosis, describe the steepness of distribution form; $\beta_K \approx 0$, it can be considered that the peak state of distribution obeys normal distribution; $\beta_K > 0$, the distribution is steep, $\beta_K < 0$, the distribution is flat.

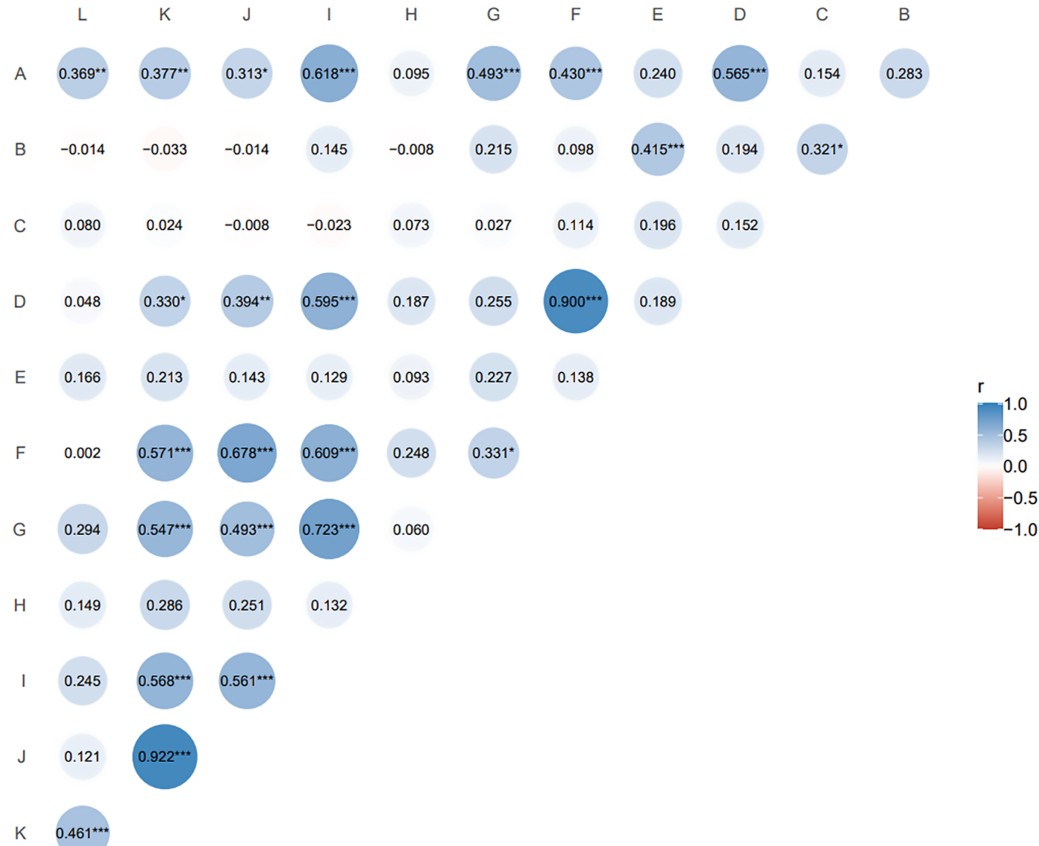

**Figure 2 Correlation analysis of agronomic traits of F1 generation.** The size and color depth of the circular image represent the degree of correlation. The larger the area of the image is, the deeper the color is, the greater the correlation between the corresponding two traits is (A) tiller height; (B) second leaf length; (C) second leaf width; (D) reproductive tiller number; (E) nutritional tiller number; (F) seed number per plant; (G) spike length; (H) spike width; (I) spikelet number per spike; (J) seed number per spike; (K) seed weight per spike; (L) thousand kernel weight. *$p \leq 0.1$, the two indexes are correlated at 0.1 level; **$p \leq 0.05$, the two indexes are significantly correlated at 0.05 level; ***$p \leq 0.01$, the two indexes are highly significantly correlated at 0.01 level; the closer the correlation coefficient ($r$) is to 1 or −1, the stronger the correlation is.

reproductive traits. All traits had a certain degree of segregation. Hybrid superiority was revealed among F1 hybrids in both nutritional and reproductive traits (Table 2).

For nutritional traits, the degree of variation ranked from largest to smallest were for the variables nutritional tiller number, second leaf length, second leaf width and tiller height, and the variation coefficient ranged from 12.67% to 34.14%. Among reproductive traits,

**Table 2 Trait heterosis analysis of F1 hybrid progeny.**

| Items | Mean | | | Variable coefficient (CV) | Mid-parent heterosis | Heterobeltiosis |
|---|---|---|---|---|---|---|
| | BOZOISKY SELECT (♂) | XJ-ALT (♀) | F1 | | | |
| Tiller height (cm) | 128.67 ± 4.22Aa | 109.40 ± 4.93Aa | 126.94 ± 4.42Aa | 12.67% | 6.64% | −1.34% |
| Nutritional tiller number | 95.00 ± 5.93Bab | 31.33 ± 5.92Cb | 171.43 ± 12.36Aa | 34.14% | 171.39% | 80.45% |
| Second leaf length (cm) | 20.70 ± 0.83Aa | 17.94 ± 1.22Aa | 22.65 ± 1.29Aa | 17.89% | 17.21% | 9.40% |
| Second leaf width (cm) | 0.34 ± 0.03Aa | 0.34 ± 0.03Aa | 0.35 ± 0.01Aa | 16.40% | 2.48% | 1.49% |
| Reproductive tiller number | 41.33 ± 1.93Bb | 17.67 ± 2.38Cc | 63.81 ± 5.55Aa | 70.18% | 116.31% | 54.39% |
| Spike length (cm) | 10.00 ± 0.15Aa | 9.11 ± 0.07Aa | 10.03 ± 0.24Aa | 15.26% | 4.90% | 0.25% |
| Spike width (cm) | 0.71 ± 0.00Aa | 0.65 ± 0.00Ab | 0.59 ± 0.02Bc | 13.24% | −12.79% | −16.27% |
| Spikelet number per spike | 34.33 ± 0.42Aa | 26.33 ± 0.16Ab | 30.93 ± 0.57Bc | 13.82% | 1.98% | −9.90% |
| Seed number per spike | 126.25 ± 3.18Aa | 41.31 ± 0.34Cc | 84.23 ± 3.83Bb | 33.30% | 0.29% | −33.49% |
| Seed number per plant (/1,000) | 3.55 ± 0.79Aa | 0.77 ± 0.08Bb | 5.87 ± 0.58Aa | 99.58% | 143.17% | 44.72% |
| Seed weight per spike (g) | 0.27 ± 0.00Aa | 0.11 ± 0.01Cc | 0.23 ± 0.02Bb | 39.22% | 18.48% | −16.65% |
| Thousand kernel weight (g) | 2.22 ± 0.02Bb | 2.00 ± 0.00Bb | 2.88 ± 0.11Aa | 15.75% | 36.80% | 30.11% |
| Means | | | | 31.90% | 47.24% (+) | 31.54% (+) |
| | | | | | −12.79% (−) | −15.53% (−) |

**Note:**
(+), The average of positive numbers; (−), The average of negative numbers. Different lowercase letters in the same line indicate significant differences at the $p < 0.05$ level, different uppercase letters in the same line indicate extremely significant differences at the $p < 0.01$ level.

seed number per plant had the highest variation level ($CV = 99.58\%$), followed by reproductive tiller number ($CV = 70.18\%$). The variation coefficients of seed weight per spike and seed number per spike were 39.22% and 33.30%, respectively, while the CV of the other reproductive traits were relatively small, with CV ranging from 13.24% to 15.75%.

The heterosis of the hybrids was statistically analyzed. Ten of the twelve phenotypic traits showed positive mid-parent heterosis, of which the nutritional tiller number showed the most obvious positive mid-parent heterosis (171.39%), followed by seed number per plant (143.17%), reproductive tiller number (116.31%) and thousand kernel weight (36.80%). Spikelet number per spike, second leaf width, spike length, seed weight per spike and second leaf length had a certain positive mid-parent heterosis, which ranged from 1.98% to 18.48%. Spike width and seed number per spike showed negative mid-parent heterosis. Seven of twelve traits showed positive heterobeltiosis. The heterobeltiosis of nutritional tiller number was the most obvious (80.45%), followed by reproductive tiller number (54.39%), seed number per plant (44.72%) and thousand kernel weight (30.11%). The heterobeltiosis of the other three traits (*i.e.*, second leaf length, second leaf width and spike length) descended from 9.40% to 0.25%. The heterobeltiosis of the other five traits was negative, ranging from 1.34% to 33.49%.

These results suggested that many traits related to seed yield in the F1 population of Russian wildrye showed obvious positive heterosis, and the nutritional tiller number related to forage yield also showed strongly transgressive segregation.

**Table 3 Principal components analysis among 12 agronomic traits of F1 *Psathyrostachys juncea*.**

| Items | $Y_1$ | $Y_2$ | $Y_3$ | $Y_4$ | $Y_5$ | $Y_6$ |
|---|---|---|---|---|---|---|
| Eigenvalue | 4.41 | 1.74 | 1.35 | 1.12 | 0.92 | 0.84 |
| Contribution rate (%) | 36.78 | 14.46 | 11.27 | 9.30 | 7.63 | 6.97 |
| Accumulative contribution rate (%) | 36.78 | 51.24 | 62.51 | 71.81 | 79.44 | 86.41 |
| Tiller height ($X_1$) | 0.32 | 0.33 | 0.03 | −0.21 | 0.32 | −0.08 |
| Reproductive tiller number ($X_2$) | 0.33 | 0.15 | −0.48 | −0.08 | 0.22 | −0.14 |
| Nutritional tiller number ($X_3$) | 0.12 | 0.36 | 0.28 | 0.30 | −0.44 | −0.41 |
| Second leaf length ($X_4$) | 0.12 | 0.58 | −0.02 | −0.01 | −0.30 | 0.07 |
| Second leaf width ($X_5$) | 0.08 | 0.42 | −0.09 | 0.31 | 0.05 | 0.77 |
| Spike length ($X_6$) | 0.31 | 0.02 | 0.30 | −0.37 | −0.16 | 0.05 |
| Spike width ($X_7$) | 0.18 | −0.04 | −0.06 | 0.62 | 0.34 | −0.37 |
| Spikelet number per spike ($X_8$) | 0.39 | −0.05 | 0.03 | −0.37 | 0.02 | −0.05 |
| Seed number per spike ($X_9$) | 0.36 | −0.35 | 0.01 | 0.19 | −0.34 | 0.17 |
| Seed weight per spike ($X_{10}$) | 0.38 | −0.31 | 0.23 | 0.22 | −0.16 | 0.17 |
| Thousand kernel weight ($X_{11}$) | 0.17 | −0.36 | 0.59 | 0.08 | 0.52 | 0.07 |
| Seed number per plant ($X_{12}$) | 0.38 | −0.12 | −0.43 | 0.07 | −0.03 | 0.01 |

## Evaluation of F1 Russian wildrye and selection of backcross parents

To explore whether multiple indicators can be combined into a few comprehensive indicators to comprehensively describe the yield and reproductive development ability of the F1 population, the related nutritional development and reproductive traits were analyzed by principal component analysis (PCA) (Table 3). Results showed that six principal component factors were identified with a cumulative contribution of 86.41%, which indicated that these six principal component factors could represent most of the variation information in the twelve observed indexes in the F1 Russian wildrye population.

The principal component factor $Y_1$ revealed 36.78% of the trait variation of F1 progeny, and the expression was $Y_1 = 0.32X_1 + 0.33X_2 + 0.12X_3 + 0.12X_4 + 0.08X_5 + 0.31X_6 + 0.18X_7 + 0.39X_8 + 0.36X_9 + 0.38X_{10} + 0.17X_{11} + 0.38X_{12}$. The principal component factor $Y_1$ was mainly determined by $X_8$ (Spikelet number per spike), $X_{10}$ (Seed weight per spike), $X_{12}$ (Seed number per plant) and $X_2$ (Reproductive tiller number). It can be considered that the first principal component represents the seed setting ability and seed yield of the F1 population.

The principal component factor $Y_2$ revealed 14.46% of the trait variation of offspring, and the expression was $Y_2 = 0.33X_1 + 0.15X_2 + 0.36X_3 + 0.58X_4 + 0.42X_5 + 0.02X_6 − 0.04X_7 − 0.05X_8 − 0.35X_9 − 0.31X_{10} − 0.36X_{11} − 0.12X_{12}$. The principal component factor $Y_2$ was mainly determined by $X_4$ (second leaf length), $X_5$ (second leaf width) and $X_3$ (nutritional tiller number). The second principal component mainly represents the tillering and leaf development ability of the F1 Russian wildrye population.

The principal component factor $Y_3$ revealed 11.27% of the variation among F1 progeny, and the expression was $Y_3 = −0.03X_1 − 0.48X_2 + 0.28X_3 − 0.02X_4 − 0.09X_5 + 0.30X_6 − 0.06X_7 + 0.03X_8 + 0.01X_9 + 0.23X_{10} + 0.59X_{11} − 0.43X_{12}$. The principal component

factor $Y_3$ was mainly determined by $X_{11}$ (thousand kernel weight), and the third principal component could explain the seed plumpness of hybrids.

The principal component factor $Y_4$ revealed 9.30% of the variation of material properties of F1 progeny, and the expression was $Y_4 = -0.21X_1 - 0.08X_2 + 0.30X_3 - 0.01X_4 + 0.31X_5 - 0.37X_6 + 0.62X_7 - 0.37X_8 + 0.19X_9 + 0.22X_{10} + 0.08X_{11} + 0.07X_{12}$. The principal component factor $Y_4$ was mainly determined by $X_7$ (spike width) and $X_5$ (second leaf width). It can be considered that the fourth principal component was mainly related to the inflorescence and leaf development of Russian wildrye.

The principal component factor $Y_5$ revealed 7.63% of the trait variation of F1 progeny, and the expression was $Y_5 = 0.32X_1 + 0.22X_2 - 0.44X_3 - 0.30X_4 + 0.05X_5 - 0.16X_6 + 0.34X_7 + 0.02X_8 - 0.34X_9 - 0.16X_{10} + 0.52X_{11} - 0.03X_{12}$. The principal component factor $Y_5$ was mainly determined by $X_{11}$ (thousand kernel weight) and $X_7$ (spike width). The fifth principal component was mainly related to the development of Russian wildrye spike, and spike width was related to seed size to a certain extent.

The principal component factor Y6 reflected 6.97% of the variation of the F1 population, and the expression was $Y_6 = -0.08X_1 - 0.14X_2 - 0.41X_3 + 0.07X_4 + 0.77X_5 + 0.05X_6 - 0.37X_7 - 0.05X_8 + 0.17X_9 + 0.17X_{10} + 0.07X_{11} + 0.01X_{12}$. The principal component factor $Y_6$ was mainly determined by $X_5$ (Second leaf width), and the sixth principal component was mainly related to the leaf development.

Through the formula $Z = (0.3678 \times Y_1 + 0.1446 \times Y_2 + 0.1127 \times Y_3 + 0.0930 \times Y_4 + 0.0763 \times Y_5 + 0.0697 \times Y_6)/0.8641$, the comprehensive score of each plant material in F1 population was calculated. It was found that there were 16 progeny plants with comprehensive scores greater than 0.80 (Table 4). Among them, the $Y_1$ score of No. 110 and No. 3 plants were significantly higher than that of other plants. $Y_1$ represents the seed setting ability of F1 Russian wildrye, and can be used as a follow-up key breeding criterion for cultivating high seed yield materials. The $Y_2$ score of the No. 4 plant was significantly higher than that of other plants. $Y_2$ represents the tillering and leaf development ability of F1 plants, which can be used as a follow-up breeding germplasm for cultivating high forage yield Russian wildrye. For No. 2, 73, 15, 67 and 1, the individual plant comprehensive performance index score in each principal component was relatively high and uniform. In this study, we finally selected No. 2 plant as the non-recurrent parent to backcross with 'BOZOISKY SELECT' Russian wildrye, and finally obtained 143 putative BC1 hybrids.

## EST-SSR primer screening

Subsequently, sixteen pairs of EST-SSR primers were randomly selected from 103 pairs of polymorphic primers to determine whether differences exist and can be stably amplified in the amplification product between the male and female parents and the hybrids. The annealing temperature had been appropriately adjusted. For the F1 population and its parents, twelve pairs of primers amplified different sequence fragments between parents and showed stable polymorphism in offspring (Table S3). Among them, five pairs of primers amplified only the paternal characteristic band, but no maternal characteristic band (6181A, 43637B, 60437, 142612A, 194938). Four primer pairs amplified the maternal characteristic band, but no paternal characteristic band (81, 45600A, 74428A, 74564).

**Table 4 Comprehensive score of traits of excellent plants in F1 *Psathyrostachys juncea*.**

| No. plant | $Y_1$ | $Y_2$ | $Y_3$ | $Y_4$ | $Y_5$ | $Y_6$ | Comprehensive score (Z) |
|---|---|---|---|---|---|---|---|
| 2 | 3.98 | 1.93 | 2.28 | 0.93 | −1.84 | −0.14 | 2.24 |
| 3 | 6.26 | −2.07 | −2.57 | 2.16 | −0.71 | 0.39 | 2.18 |
| 110 | 6.58 | −2.14 | −5.09 | 0.24 | 1.14 | 1.51 | 2.03 |
| 73 | 2.31 | 0.93 | 1.61 | 1.86 | 0.36 | 1.82 | 1.73 |
| 7 | 4.51 | −1.09 | 0.43 | 0.65 | −0.61 | −1.21 | 1.71 |
| 4 | 2.02 | 5.17 | −2.74 | 3.29 | −0.56 | −1.03 | 1.59 |
| 91 | 2.36 | 2.89 | −1.32 | −0.26 | −0.34 | 3.54 | 1.54 |
| 9 | 3.64 | −1.38 | 1.20 | 0.74 | −0.68 | 0.21 | 1.51 |
| 15 | 2.17 | 0.55 | 1.92 | 1.74 | −0.03 | −1.23 | 1.35 |
| 67 | 2.03 | 0.99 | 0.17 | 0.16 | 1.74 | 0.07 | 1.23 |
| 111 | 3.48 | −0.80 | −1.60 | 0.06 | 1.42 | −0.53 | 1.23 |
| 1 | 4.06 | 0.75 | −1.86 | −0.51 | −2.03 | −2.37 | 1.19 |
| 123 | 2.69 | −0.63 | 1.07 | −1.00 | −0.19 | 1.57 | 1.18 |
| 81 | 2.03 | 0.39 | 1.60 | −0.46 | 0.30 | −0.21 | 1.10 |
| 16 | 2.93 | −1.71 | 0.87 | 0.72 | −1.32 | 0.19 | 1.05 |
| 38 | 2.82 | −3.05 | 1.05 | −0.48 | −1.14 | 1.74 | 0.81 |

Three pairs of primers amplified both the paternal and maternal characteristic bands (28553, 44262, 51628) (Fig. 3), and they were used to quickly identify the true F1 hybrids. For BC1 population identification, eight pairs of primers amplified different sequence fragments between parents. Among them, two primer pairs only amplified the paternal characteristic band (43637B, 74564). Three primer pairs amplified the maternal characteristic band (81, 28553, 74428A). Three primer pairs amplified both the paternal and maternal characteristic bands (6181A, 44262, 60437) (Fig. 3), and they were used to quickly identify the purity of the BC1 population.

**Hybrid identification**

Three EST-SSR primer pairs (28553, 44262 and 51628) with stable parent characteristic markers were selected to identify the authenticity of the F1 hybrid population of Russian wildrye (Table 5). Three EST-SSR primer pairs (6181A, 44262 and 60437) were selected to identify the authenticity of the BC1 population of Russian wildrye (Table 6). According to the SSR markers, the differences between 123 F1 offspring plants and their parents were compared. It was shown that plants with one or more characteristic markers the same as the male parent were true hybrids. The results showed that among 123 Russian wildrye F1 hybrids, 119 were true hybrids, and four were false hybrids without paternal characteristic band (numbered 43, 57, 86 and 105). Among 143 Russian wildrye BC1 hybrids, 137 were true hybrids, and six were false hybrids (numbered 28, 43, 75, 113, 128 and 139).

Hybrid purity (%) (F1) = (123−4)/123 × 100% = 96.75%

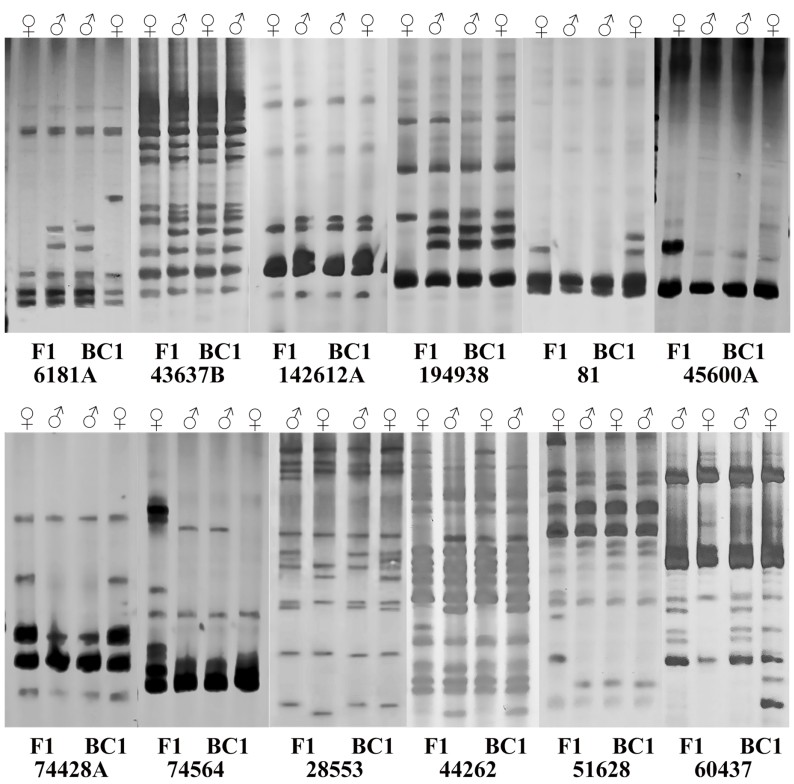

**Figure 3 Polymorphism amplification of different EST-SSR primers in parents of F1 and BC1 populations.**

**Table 5 EST-SSR primer sequences and statistical analysis of amplification results in F1 generation.**

| Primer pair | Sequence (5′–3′) Forward/reverse | na* | ne* | h* | I* | AT/°C | T | PPB/% | FF1 | MF1 |
|---|---|---|---|---|---|---|---|---|---|---|
| 28553 | TTGACAAATCAGTCAGGCCA/ ACACCGAGAAATCCCATCAC | 2.0000 | 1.7322 | 0.4117 | 0.5989 | 54.5 | 10 | 100.00% | 2 | 3 |
| 44262 | TACGACTTCCTCGAACACCC/ GTCCAGTCGTCGATCTCCTC | 1.8125 | 1.6136 | 0.3370 | 0.4873 | 57.0 | 16 | 81.25% | 2 | 3 |
| 51628 | CTTAAGTTGCATGTCCCCGT/ TGGTCCATTCTCTTGGGAAG | 1.9167 | 1.6866 | 0.3745 | 0.5424 | 54.5 | 12 | 91.67% | 3 | 2 |
| Mean | | 1.9167 | 1.6775 | 0.3744 | 0.5429 | 55.33 | 12.67 | 90.97% | 2.33 | 2.67 |

**Note:**
na*, observed number of alleles; ne*, effective number of alleles (*Kimura & Crow (1964)*); h*, *Nei's (1973)* gene diversity; I*, Shannon's Information index (*Lewontin (1972)*); AT, optimized annealing temperature/°C ; T, total number of amplified bands; PPB, percentage of polymorphic bands/%; FF1, female characteristic bands; MF1, male characteristic bands.

$$\text{Hybrid purity } (\%) \, (\text{BC1}) = (143 - 137)/143 \times 100\% = 95.80\%$$

Gel electrophoresis assay of Russian wildrye hybrid lines and their parents using EST-SSR marker 44262 are shown in Fig. 4 (F1) and Fig. 5 (BC1).

Hybrid purity was further verified by self-fertility testing through bagging isolation. The average self-fertility rate of the female parent single spike was 2.45%, with a range from 1.79% to 3.76% (Table 7). The proportion of false hybrids in the F1 population identified by SSR markers was 4/123 = 3.25%, which was within the self-fertility rate range.

**Table 6 EST-SSR primer sequences and statistical analysis of amplification results in BC1generation.**

| Primer pair | Sequence (5′-3′) Forward/reverse | na* | ne* | h* | I* | AT/°C | T | PPB/% | FF1 | MF1 |
|---|---|---|---|---|---|---|---|---|---|---|
| 6181A | CCTTTTCCGTGCATACTGGT/ CTGCGAGGGAATGATGGTAT | 1.5000 | 1.4335 | 0.2311 | 0.3272 | 55.0 | 6 | 50.00% | 1 | 2 |
| 44262 | TACGACTTCCTCGAACACCC/ GTCCAGTCGTCGATCTCCTC | 1.4000 | 1.2518 | 0.1464 | 0.2178 | 57.0 | 15 | 40.00% | 1 | 2 |
| 60437 | GAAGAACAGGGACTGGACGA/ TGGGGAAGAGTCTCACTTGG | 1.8000 | 1.5622 | 0.3251 | 0.4758 | 56.7 | 10 | 80.00% | 2 | 2 |
| Mean | | 1.6000 | 1.4335 | 0.1464 | 0.3403 | 56.2 | 10.33 | 56.67% | 1.33 | 2 |

**Note:**
na*, observed number of alleles; ne*, effective number of alleles (*Kimura & Crow (1964)*); h*, *Nei's (1973)* gene diversity; I*, Shannon's Information index (*Lewontin (1972)*); AT, optimized annealing temperature/°C ; T, total number of amplified bands; PPB, percentage of polymorphic bands/%; FF1, female characteristic bands; MF1, male characteristic bands.

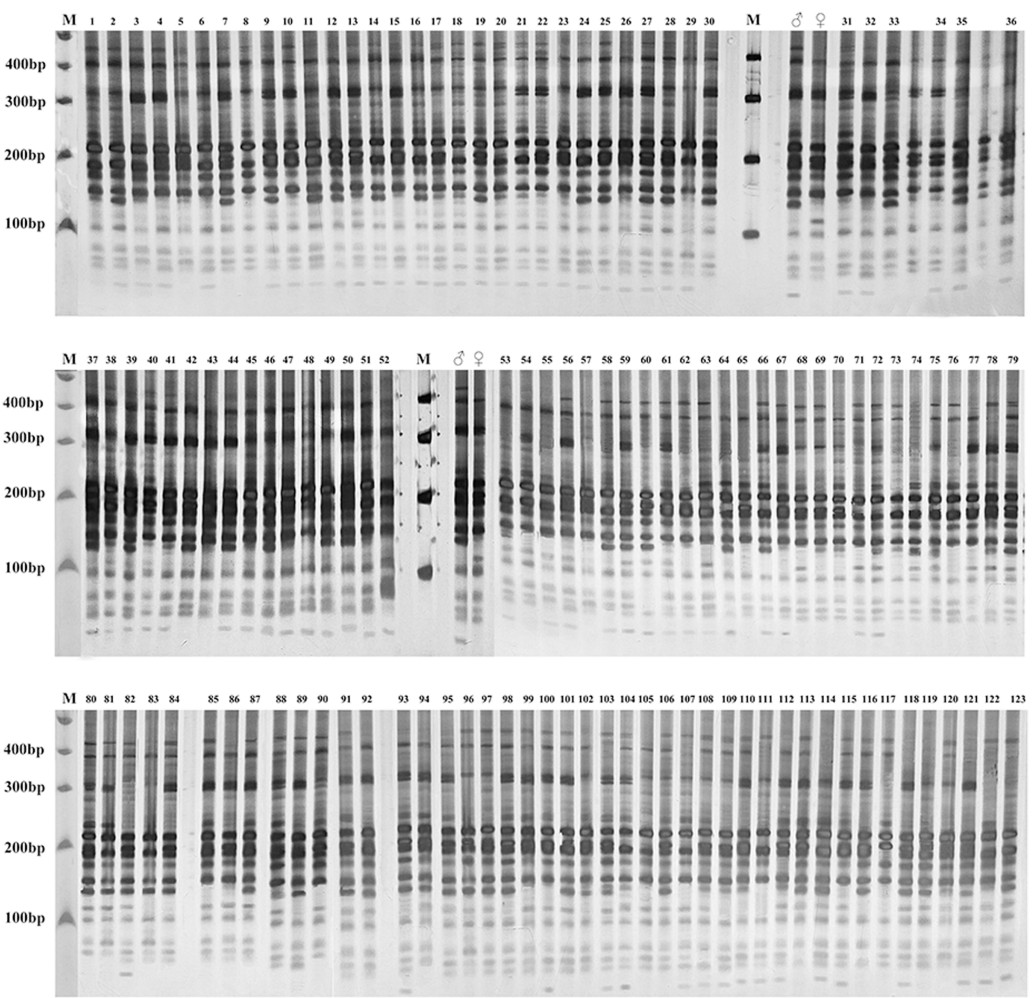

**Figure 4 Gel electrophoresis assay of Russian Wildrye F1 hybrids and their parents (EST-SSR primer 44262).** M = marker size; ♀ = the female parent; ♂ = the male parent; 1–123, identified plants.

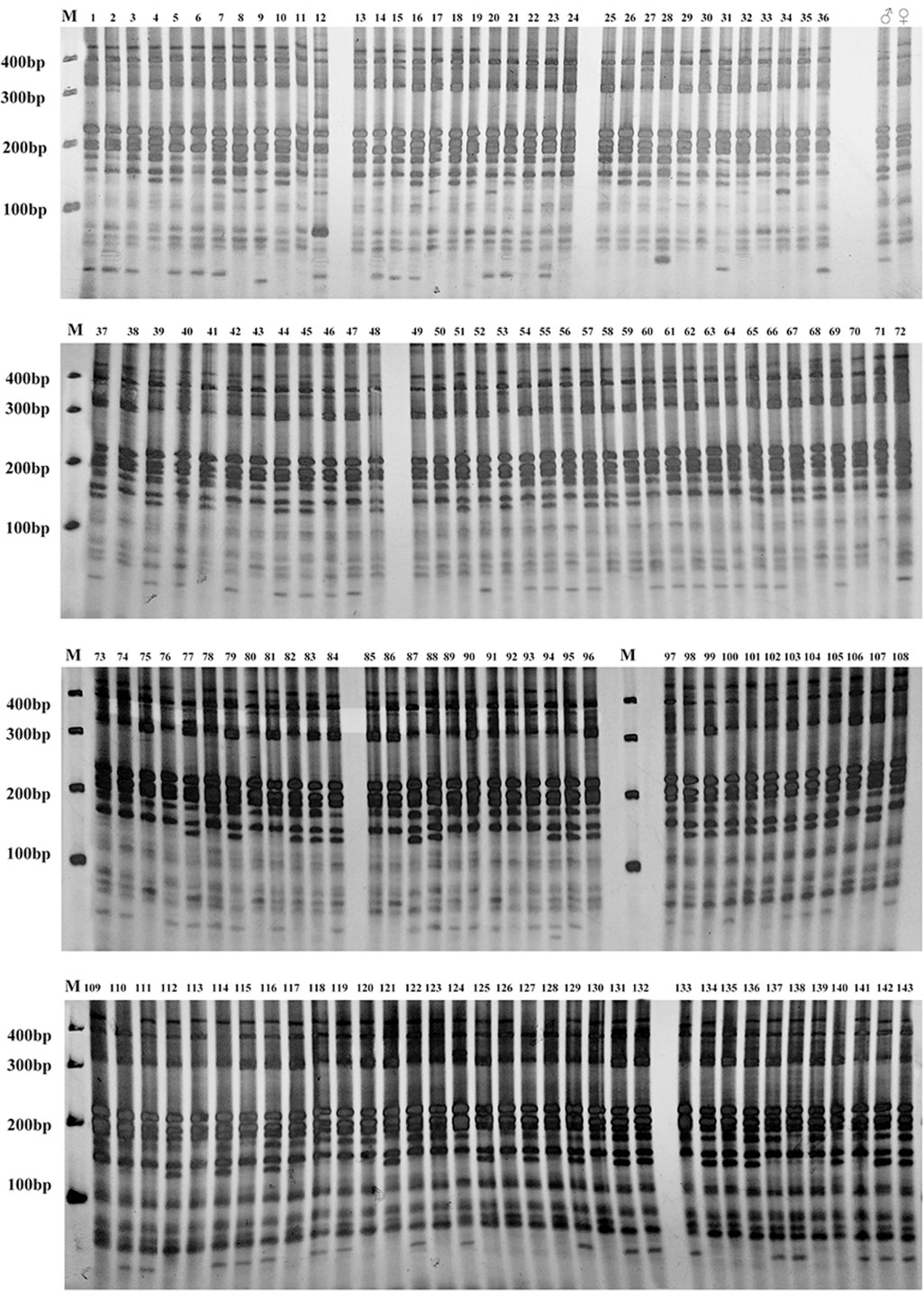

**Figure 5 Gel electrophoresis assay of Russian wildrye BC1 hybrids and their parents (EST-SSR primer 44262).** M = marker size; ♀ = the female parent; ♂ = the male parent; 1–143, identified plants.

**Table 7 Comparision of self-fertility rate between F1 generation and the female parent.**

| Items | Self-fertility rate (%) | | | Standard deviation (SD) | Significance level ($p \leq 0.05$) | Extremely significant level ($p \leq 0.01$) |
|---|---|---|---|---|---|---|
| | Max | Min | Mean | | | |
| F1 individual plant | 1.13% | 0.32% | 0.64% | 0.0031 | b | B |
| F1 single spike | 0.00% | 0.64A% | 0.25% | 0.0034 | b | B |
| Female parent single spike | 3.76% | 1.79% | 2.45% | 0.0083 | a | A |

Note:
Compared with the same column, the lowercase letter showing significant difference ($p \leq 0.05$) and uppercase letters showing extremely significant difference ($p \leq 0.01$).

The hybrid identification results were further confirmed. The self-fertility rate of the female parent was significantly higher than that of the F1 population, and there was no significant difference between the self-fertility rate of F1 individual plants and single spikes. However, it was found by observation that the self-fertility rate of different F1 individual plants was quite different, with a maximum self-fertility rate of 1.13% and a minimum of 0.32%. Single spike of F1 hybrid plants under bagging isolation was nearly completely sterile. The proportion of false hybrids in the BC1 population identified was 4.20% which was slightly beyond the tested self-fertility rate range of the female parent plant. This suggested that the self-incompatibility level varied among individuals of Russian wildrye under bagging isolation.

## SSR-PCR polymorphism analysis and genetic similarity analysis of hybrids

The amplified fragments of the three primer pairs (28553, 44262 and 51628) for 119 true F1 hybrids and their parents were different in size and varied between 40 and 550 bp. There were 38 clearly identifiable markers. The total number of markers amplified by each primer pair ranged from ten to sixteen, and the percentage of polymorphism varied from 81.25% to 100.00%, with an average of 90.97%. Each primer pair could generate two or three paternal characteristic bands (Table 5).

The amplified fragments of the three primer pairs (6181A, 44262 and 60437) for 137 true BC1 hybrids and their parents were different in size and varied between 40 and 550 bp. The optimal annealing temperature and amplification results of the primers were arranged. There were 31 clearly identifiable markers. The total number of markers amplified by each primer pair ranged from six to fifteen, and the percentage of polymorphism varied from 40.00% to 80.00%, with an average of 56.67%. Each primer pair could generate two paternal characteristic bands (Table 6).

Using the amplified markers of true hybrids and their parents as the original data matrix, genetic similarity coefficients were analyzed. The genetic similarity coefficients of true F1 hybrids and their parents ranged from 0.474 to 0.974, and was 0.605 between male and female parents. The genetic similarity coefficients of hybrids No. 37 and No. 39 were the largest (0.974), while those of hybrids No. 1 and No. 12 or No. 65 and No. 118 were the smallest (0.474). The genetic similarity coefficients between the progeny and the male parent ranged from 0.500 to 0.895, and those between the progeny and the female parent

ranged from 0.526 to 0.868 (Table S4). The genetic similarity coefficients of true BC1 hybrids and their parents ranged from 0.606 to 1.000, and was 0.697 between male and female parents. The genetic similarity coefficients between the progeny and the male parent ranged from 0.667 to 0.939, and those between the progeny and the female parent ranged from 0.667 to 0.909 (Table S5).

## DISCUSSION

### Parent selection and evaluation of agronomic traits of the hybrid population

In all hybrid combinations in this research, the nutritional tiller number and reproductive tiller number of male parent group were significantly higher than those of the female parent group. However, the female parent group has some advantages in seed traits. *Bai (2016)* found out that the average spike length of the female parent group was 47.62% longer than that of the male parent group, and the spikelet number per spike of the female parent group was 35.29% higher than that of the male parent group. The seed setting rate of 'XJ-ALT' was highest among all female parents of fifteen hybrid combinations. Germplasm trait evaluation showed that the reproductive tiller number of 'BOZOISKY SELECT' reached 113, while that of 'XJ-ALT' was only 28 in the second year of growth. Thus, the cross combination conforms to the principle of parental selection.

Russian wildrye is a cross-pollinated plant with self-incompatibility and the genotype of the parents were not homozygous. The major gene plus polygene mixed inheritance model could be used to further analyze the inheritance of F1 population, calculate the inheritability of each trait, and find out whether the traits can be stably inherited in future generations. This strategy has also been widely used in forage breeding (*Dong et al., 2021*). The major gene plus polygene mixed inheritance model analysis method for quantitative traits has been used to analyze the genetic model of Triticale spike length, spikelet number and seed number per spike. *Chang et al. (2021)* analyzed Triticale and found that the skewness and kurtosis of spike length, spike number and seed number per spike were between 0 and 1, and these traits were approximately in line with the normal distribution, indicating that these traits may be co-controlled by the main gene and polygenes. In our study, the distribution of seed number per spike basically conformed to a normal distribution and the skewness and kurtosis were between 0 and 1 in the Russian wildrye F1 population, indicating that the inheritance of seed number per spike was under polygenic control. The skewness and kurtosis of the second leaf length, second leaf width, nutritional tiller number and seed number per spike were all greater than 1. This suggests that there may be a certain level of interaction effect between genes regulating these quantitative traits.

The purpose of making a backcross between the F1 Russian wildrye and the bred variety "Bozoisky-Select" was to accumulate the high biomass yield genes related to dense tiller number of "Bozoisky-Select", and integrate high seed yield genes related to greater spikelet number into the genetic background of wild Russian wildrye on the basis of retaining the drought and cold resistance of wild germplasm. The overwintering rate of the F1

population was 83.67%. The hybrid plant population entered the peak reproductive and nutritional growth period in the fourth year, which was consistent with reports on Russian wildrye germplasm introduction (*Yun et al., 2006*; *Wang et al., 2014*; *Bai, 2016*). After analyzing the heterosis of the hybrid progeny of Russian wildrye, it was found that the F1 population had obvious heterosis in vegetative growth and heterobeltiosis of nutritional tiller number (80.45%), which was more obvious than that of reproductive growth and heterobeltiosis of reproductive tiller number (54.39%). Seed weight of F1 Russian wildrye was higher than that of its parents. These results suggest that the positive development F1 Russian wildrye forage production traits and the potential occurrence of a heterotic response regarding biomass productivity.

In this study, each plant of the F1 population was further evaluated using PCA-synthesis score calculation. In perennial forage grass breeding, there should be a balance between the nutritional traits for forage production and reproductive traits for seed production. Multiple traits need to be considered to identify target hybrid plants derived from cross-pollinated parents. The PCA-synthesis score method provided an effective way for backcross parent selection of this perennial forage grass species.

## Efficient utilization of EST-SSR in hybrid identification

SSR, as the co-dominant and neutral molecular markers, has been widely used in hybrid offspring identification and provides a theoretical basis for subsequent research. The results of SSR markers can be verified by PCR, and the working cycle can be greatly shortened (*Nadeem et al., 2014*; *Li et al., 2019*; *Zhong et al., 2021*). EST-SSR is the identification of SSR by electronic screening using existing EST sequences, followed by PCR detection (*Zhang et al., 2020*). Using EST to develop SSR avoids the cloning and sequencing steps in the process of developing SSR primers, makes full use of existing data, and reduces development costs (*Fan et al., 2020*). EST-SSR is well conserved and has good versatility among different species, and can distinguish materials with relatively close genetic relationships (*Zhang & Tang, 2010*). For example, nine EST-SSR primers from different sources were successfully amplified for *Elymus sibiricus* hybridization identification and EST-SSR markers have been developed from *Elymus wawawaiensis*, *Elymus lanceolatusd*, *Pseudoroegeneria spicata* and *Leymus* species (*Zhao et al., 2017*). The EST-SSR primers used in this study were picked from expressed sequences related to tiller traits, and tiller number was the most significant impact factor on biomass yield and seed yield of Russian wildrye (*Zhang et al., 2018*). EST-SSR markers can not only distinguish genotypes of hybrid progeny, but is also useful for a hybrid breeding procedure related to tiller trait improvement.

Hybridization was made using parents with significant differences in tiller number trait. Among sixteen pairs of primers used to test the amplifying difference between parents, twelve primer pairs amplified different products between parents of the F1 population, and eight pairs of primers amplified different products between parents of the BC1 population. In total, five EST-SSR primer pairs with stable parent characteristic markers were selected to identify the authenticity of the F1 and BC1 hybrid populations of Russian wildrye, including one primer pair shared in both populations. Identification results of both F1 and

BC1 populations suggested that the EST-SSR molecular markers developed from the genome sequence of Russian wildrye that were especially related to expressed gene locus could effectively reveal the genetic differences between parents and offspring. The results of true hybrid identification by EST-SSR were confirmed with a self-fertility test by bag isolation. The self-fertility test showed that the self-fertility ability varied between individual plants of Russian wildrye. For some plants, self-fertility could occur between flowers of the same spike in a low frequency (1.79–3.76%), but this rarely happened for some other single plants.

The genetic similarity coefficients of true F1 hybrids and their parents ranged from 0.474 to 0.974, and was 0.605 between male and female parents. The genetic similarity coefficients of true BC1 hybrids and their parents ranged from 0.606 to 1.000, and was 0.697 between male and female parents. The genetic similarity coefficients between the F1 progeny and the male parent ranged from 0.500 to 0.895, and those between the BC1 progeny and the male parent ranged from 0.667 to 0.939. The individual genetic difference of the backcross population was significantly less than that of the F1 population. This result was in line with the genetic law of backcrossing and suggested that a Russian wildrye population inclined to a recurrent parent was obtained. A subset of individuals in the backcross population had closer genetic background to the recurrent parent, but there was still a high level of genetic variation in the BC1 population. Further evaluation and selection of the BC1 population is still needed.

The EST-SSRs developed in this research also provide theoretical tools for correlation analysis between tiller traits and molecular markers, and could play an important role in future breeding programs to increase tiller number and forage yield. The backcross population and EST-SSRs could be used in molecular genetic map construction and QTL mapping for tiller number and other important traits.

## CONCLUSIONS

The authenticity of Russian wildrye hybrids can be identified efficiently and accurately by EST-SSR markers developed for the Russian wildrye genome. Three primer pairs that amplified both the paternal and maternal characteristic bands were used for F1 population identification, and three primer pairs (including one shared primer pair) were used for BC1 population identification. The hybrid purity was 96.75% for the F1 population and 95.80% for the BC1 population. The results were further verified by measuring the maternal self-fertility rate. Furthermore, the coefficient of variation (CV) of the 12 phenotypic traits of the hybrids ranged from 12.67% to 99.58% showed that the F1 population showed obvious segregation. Nutritional tiller number, second leaf length, reproductive tiller number, seed number per plant, thousand kernel weight showed both mid-parent heterosis (mean = 98.00%) and heterobeltiosis (mean = 50.05%), indicating that this population can exhibit heterosis in forage yield and seed yield. Fertility of the backcross population was obtained using the high-quality parent "Bozoisky-Select" as recurrent parent, which could be used as breeding material for further study. The genetic similarity coefficients between the F1 progeny and the male parent ranged from 0.500 to 0.895, and those between the BC1 progeny and the male parent ranged from 0.667 to 0.939.

A subset of individuals in the BC1 population had closer genetic distance to the recurrent parent, and genetic variation within the BC1 population decreased compared to the F1 population.

## ACKNOWLEDGEMENTS

We thank the test support staff in the Ministry of Education Key Laboratory of Grassland Resources, Inner Mongolia Agricultural University, and are grateful for the material support of the Forage and Range Research Laboratory of USDA (USDA-ARS-FRRL), the U.S. National Plant Germplasm System (NPGS) and the China National Medium-term Gene Bank of Forage Germplasm.

### Funding

This work was supported by the National Natural Science Foundation of China (No. 31860672). The funders had no role in study design, data collection and analysis, decision to publish, or preparation of the manuscript.

### Grant Disclosures

The following grant information was disclosed by the authors:
National Natural Science Foundation of China: 31860672.

### Competing Interests

The authors declare that they have no competing interests.

### Author Contributions

- Zhiqi Gao conceived and designed the experiments, performed the experiments, analyzed the data, prepared figures and/or tables, authored or reviewed drafts of the article, and approved the final draft.
- Lan Yun conceived and designed the experiments, performed the experiments, analyzed the data, prepared figures and/or tables, authored or reviewed drafts of the article, and approved the final draft.
- Zhen Li performed the experiments, analyzed the data, prepared figures and/or tables, and approved the final draft.
- Qiyu Liu performed the experiments, prepared figures and/or tables, and approved the final draft.
- Chen Zhang performed the experiments, prepared figures and/or tables, and approved the final draft.
- Yingmei Ma performed the experiments, authored or reviewed drafts of the article, and approved the final draft.
- Fengling Shi analyzed the data, authored or reviewed drafts of the article, and approved the final draft.

## Data Availability

The code and raw data are available in the Supplemental Files.

## Supplemental Information

Supplemental information for this article can be found online at http://dx.doi.org/10.7717/peerj.14442#supplemental-information.

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
