# Peer review of "Hybrid purity identification using EST-SSR markers and heterosis analysis of quantitative traits of Russian wildrye"

_PeerJ, doi:10.7717/peerj.14442_

## Round 0.1 · original submission · Major Revisions

According to reviewers comments, this paper cannot be accepted for publication. It needs a major revision in order to be reconsidered for publication. In particular, results are not properly presented and additional statistics are necessary to corroborate overall findings on heterosis. Moreover, several parts of the paper are not scientifically sound and need to be rewritten, including the Introduction section. Authors are invited to revise the paper taking into careful consideration all the suggestions made by the reviewers. Please note that requested changes are required for publication.

Reviewer 1 ·

Basic reporting

The manuscript is not written in professional English and it is not suitable for international readership. There is a great deal of superfluous information in the Introduction and Discussion. The authors did not formulate scientifically or agriculturally meaningful objectives. The Introduction should focus on the botany and agricultural uses of Russian wildrye, which could be of interest and usefulness to many people. Markers are useful tools, but they are not scientifically interesting at this point. The authors constructed a potentially useful EST library for Russian wildyre, which is useful but the technology and approaches used for the DNA markers is not new or interesting. I suggest that the authors discuss potential applications of the Russian wildrye EST library, but otherwise eliminate text related to DNA from the Introduction and Discussion. This article is not the place to review DNA markers in other species unless those species are very closely related. Likewise, we don't care about the genome designation unless you are discussing genome designations of closely related species.

Does this journal really allow references to be given with OneName et al.? I know some journals allow lengthy lists to be shortened, but this does not seem professional to me.

Experimental design

I think that the research is basically within the scope of many high quality journals, but meaningful research questions were not well defined and the study is incomplete without a linkage map and QTL analysis. Results of this study would be easier to interpret if plants were clonally replicated in the field. It is still possible to conduct a meaningful QTL analysis with these data, but comparisons of trait values among individual plants is meaningless without replication.

Validity of the findings

Results of this study would be easier to interpret if plants were clonally replicated in the field. It is still possible to conduct a meaningful QTL analysis with these data, but comparisons of trait values among individual plants is meaningless without replication.

In my view, this study is incomplete. The authors need to genotype at least 500 polymorphic markers, develop linkage maps, and conduct a QTL analysis. This looks like a preliminary report that could be given in a workshop.

Development of the EST library for Russian wildrye has many potential applications and this alone could constitute a short paper. However, it is not clear to me if the authors have made the DNA sequences publicly available. The authors included a few primer sequences in Table 3, but this is not a major contribution to the body of scientific literature. The validity of the DNA markers should have been tested to see if they fit expected segregation ratios of 3:1, 1:1, or possibly 1:2:1 (if they think they have a codominant marker).

The conclusion that "heterosis is obvious" is NOT supported by the data. Table 5 is probably the most interesting data in the paper. However, it is difficult to interpret because the parental means and F1 progeny means to not have standard errors because they were not replicated. What does column labeled P mean? Is this the mean of both parents. Ideally, the authors would report means and standard errors for EACH parent, so we know if parents are different. I don't think it is really possible to conduct meaningful statistical analysis of these data. If it is possible, the authors failed to make it clear and demonstrable.

Additional comments

The authors have obviously done some good and useful work, but it really seems like this is a preliminary report. I think most readers would expect at least one or all of four possible things: 1) public release of all data including entire EST library, 2) a linkage map, 3) QTL analysis, and 4) heritability of each trait. It would be possible to determine heritability of traits if the authors can construct a genetic similarity matrix of plants with good genome coverage. All of these things would be interesting and useful. Demonstration of heterosis would also be interesting, but it is not clear if the authors have statistically sound data to prove it.

Annotated reviews are not available for download in order to protect the identity of reviewers who chose to remain anonymous.

Reviewer 2 ·

Basic reporting

authors should improve the writing of this manuscript

Experimental design

no comment

Validity of the findings

conclusion section should be rewritten

Additional comments

This manuscript describes develop SSR markers, identify true hybrids and analysis genetic and agronomic traits of hybrid population in Russian wildrye. These data are solid, and this work is worth to be published. 
However, authors should improve the writing of this manuscript. For example, the Abstract section needs to be shorted, and the reference format needs to be rewritten (many references lack of pages), and Acknowledgments section lack of Project funding information, and conclusions section also should rewritten. Other issues:
1.In Line 247, “29889” should be “29,889”
2.In line 259, four pairs of primers should be verify again by sequencing of PCR product.
3.In line 280, “it can be seen from Table 4” should be “(Table 4)”
4.In line 291, “1300bp” should be “300bp”
5.In line 315 to 317, hard to understand
6.In line 378, “in recent years, there have been many reports in this field” should be “With the development of .., ... has been widely used to ... in many plants ”.
7.In line 399 and 404, Two “For example” are applied in one paragraph.
8.In line 453 and 472, “in this study” is used at the beginning of each paragraph, how to reflect the beauty of the language.
9.In line 486, a spaces appears.
10.In line 497 and 498, should be rewritten.
11.In line 499 to 506, A general description is required.
12.Figure 2, should change to original gel image.

---

## Round 0.2 · Major Revisions

Your manuscript has now been reviewed by experts in the field and it still requires major compulsory revisions. Please revise the manuscript very carefully according to each of the comments made by the reviewers and upload the revised file within 3 weeks. Please note that the manuscript suffers from significant flaws and needs to be improved for its form and contents. Reviewers reported that important elements of the Discussion are not adequately supported by the Results. Please revise this section accordingly.

Reviewer 1 ·

Basic reporting

The article is not structured very well and the English grammar needs improvement. The abstract failed to summarize major findings related to trait differences, trait correlations, and heterosis.

The Introduction did not have well formulated objectives.

The results did not elucidate potentially interesting and important trait differences between the parents (Table 5 did not even define P1 and P2). such as:

1) "P1" showed significantly greater nutritional tiller number, reproductive tiller number, seeds per spike, seeds per plant, seed yield per spike compared to "P2"
2) "P2" did not show any significantly better traits compared to "P1"
3) Compared to "P1" and "P2", the hybrid population showed significantly greater nutritional tiller number, reproductive tiller number, and thousand kernal weights.

It is frustrating that P1 and P2 are undefined in Table 5? WHY IS THAT? The authors also refer to male and female parents, but the authors did not test for reciprocal differences so this is really irrelevant. Being the male parent of female parent is not important unless you can show that there are reciprocal differences when both parents are used as male and female parents (i.e. maybe there are cytoplasmic effects of the female parent). Please use the names of the parents (e.g. "Bozoisky Select" or "XJ-ALT") consistently throughout the entire manuscript (not "male" or "female" or "P1" or "P2"). It creates unnecessary confusion.

Important elements of the Discussion are not supported by the Results. For example, the authors make argument that both parents have useful traits, but this was not supported by results in Table 5.


The authors often refer to "previous studies" without citation.

Experimental design

The author did not adequately describe tests used to identify "differentially expressed genes" (DEGS). I think that should be subject of a different paper. It is not necessary for this paper. I think is is sufficient to say that tissues used to me the EST library come from plants with high tiller number and low tiller number.

Validity of the findings

Important elements of the Discussion are not supported by the Results. For example, the authors make argument that both parents have useful traits, but this was not supported by results in Table 5. The authors should discuss if the results of this study agree with other studies.

In the Discussion, the authors state that "After transplanting in autumn, the overwintering rate of the F1 population was 83.67%." Is there a citation for this? If not, then this should be reported in the Results.

In the Discussion, the authors state that "Therefore, it was suggested that more markers should be selected to accurately evaluate the genetic diversity of hybrids." Are you saying that you did not use enough markers?

Annotated reviews are not available for download in order to protect the identity of reviewers who chose to remain anonymous.

Reviewer 2 ·

Basic reporting

The authors have extensively revised this manuscript since an earlier submission and it is greatly improved. The article is not well written and the study suffers from significant flaws. And writting is the major problem.
1. The abstract is too long. This abstract is not a list of the experimental results
2. the innovation and significance of this study are not clearly and well writted

Experimental design

no comment

Validity of the findings

The novelty and the conclusion are not well stated.

Additional comments

1. The author should modify the language well first.
2. The novelty should clearly written in introduction section. I'm hardly to find.
3. More time should be take and rewritten the paper.
eg. line 23, 36, 64, 98, 465, 452,465

Annotated reviews are not available for download in order to protect the identity of reviewers who chose to remain anonymous.

---

## Round 0.3 · Minor Revisions

As you can see, the final comments from the reviewers relate to improving the language and grammar of the article. Please address these comments in a final revision.

Reviewer 1 ·

Basic reporting

The manuscript is much improved, but it still needs extensive professional editing for English grammar and scientific content. For example the second sentence of the abstract begins "As one of high-quality grazing forage...", is incorrect grammar. Perhaps you could say "As a high-quality forage..." The term "high-quality" is relative to what? This kind of description it not really helpful. Russian wildrye might be one of the best available forage grasses in some areas, but it is certainly not among the highest-quality forages compared to alfalfa or perennial ryegrass. "Quality" is relative, so it only has meaning if you can compare it something or put it into context of something meaningful.

The abstract contains excessive details and information that is not important for general readers such as "All traits had a certain degree of segregation", where "certain degree" could mean anything or nothing.

Need to remove fluffy adjectives like "great" or "excellent". It would be more helpful if you can cite literature that compares species. A good abstract should be under 250 words.

Experimental design

The experimental design is basically sound, with high technical standards. The manuscript describes original research that is within the Aims and Scope of the journal.

Validity of the findings

Findings appear to be statistically sound and controlled. The conclusions are supported by the results. I do not see any inappropriate claims of impact or novelty.

Reviewer 2 ·

Basic reporting

1. The abstract section needs to be reduced length. Recommend writing 8-10 sentences.
2. The conclusion part needs to summarize the results. I suggest 3-4 sentences are enough.
3. In introduction section, line 80 should add content about the importance of hybrid research in Russian wildrye.

Experimental design

no comment

Validity of the findings

no comment

Additional comments

At last, the English grammar needs improvement.

---

## Round 0.4 · accepted · Accept

Dear Authors

I am pleased to inform you that after the last round of revision, the manuscript has been improved a lot, and it can be accepted for publication.

Congratulations on the acceptance of your manuscript, and thank you for your interest in submitting your work to PeerJ

Best Regards